# Audience segmentation of New Zealand cat owners: Understanding the barriers and drivers of cat containment behavior

**Sarah A. E. Chamberlain**, **Lynette J. McLeod**, **Donald W. Hine**\*

School of Psychology, Speech and Hearing, University of Canterbury, Christchurch, New Zealand

\* donald.hine@canterbury.ac.nz

**Data Availability Statement:** The data file is available from the Mendeley database (DOI: 10.17632/hytxh9bm8c.1).

## Abstract

Free-roaming companion cats have a detrimental impact on the environment and are at risk of harm. Despite these negative impacts, it is the norm in New Zealand (NZ) to allow companion cats to roam freely and only a minority of cat owners practice cat containment. This study firstly sought to identify what factors act as barriers and drivers of NZ owners' participation in cat containment, and secondly whether NZ owners could be segmented into unique audiences based on the factors predicting their cat containment behavior. It was hypothesized that cat owners with greater capability, opportunity, and motivation to perform cat containment would have greater cat containment intentions and behavior. Furthermore, it was expected that at least three segments of cat owners would exist in NZ which differed significantly in the set of capability, opportunity and motivational factors predicting their cat containment behavior. A quantitative online cross-sectional survey of 395 NZ cat owners was conducted, measuring containment intentions and behavior, and capability, opportunity, and motivation to perform cat containment. Results from bivariate correlations and multiple regression demonstrated that capability, opportunity, and motivational factors predicted increased cat containment intentions and behavior. Latent profile analysis identified four distinct segments of cat owners with unique COM profiles; *engaged* (6%), *receptive* (17%), *ambivalent* (48%), and *opposed* (30%). Validation analysis demonstrated that these groups all differed significantly in their cat containment intentions and behaviors. From these findings theoretically grounded behavior change interventions can be developed to target the causes of non-participation in cat containment for each of the identified cat owner segments, thereby improving the management of free-roaming cats in NZ.

## Introduction

The domestic cat, (*Felis catus*), is New Zealand's (NZ) most popular companion animal with an estimated population of 1.2 million [1, 2]. Companion cats in NZ are defined as common domestic cats that reside with humans and depend on them for their welfare [2]. Many owners choose to have a companion cat because of the perceived friendship, love, and affection they offer. The negative effect companion cats have on the environment is, however, seemingly overlooked by NZ owners. Free-roaming cats are considered to have several negative

**Funding:** The authors received no specific funding for this work.

environmental impacts [3, 4], with one of the most notable being wildlife predation [5–9]. While evidence is limited as to the exact impact companion cats have on wildlife through predation [3, 4], research using owners' reports of prey brought home and animal-borne camera tracking supports that companion cats are prolific hunters in both NZ [5–7] and international contexts [8, 9]. In addition to predation, free-roaming cats are implicated in other ecological effects, such as disease transmission and behavioral changes (e.g., breeding behavior, parental care, and stress induction) [10]. The spread of *Toxoplasma gondii* by free roaming cats is of particular concern [4, 11, 12]. This parasite causes Toxoplasmosis, an infection that has been found it all regions globally, which poses a risk to the health of humans, domestic animals and wild animals [4, 11, 12]. The welfare of cats allowed to free roam is also at risk due to their increased likelihood of injury, illness, and death [4, 8, 13]. Using animal-borne camera tracking, research in NZ and the United States has found that companion cats frequently wandered on the road, had altercations with other cats, and consumed foreign substances [8, 13]. Furthermore, Elliot and colleagues [14] found that over two thirds of NZ owners had lost a cat to roaming-related causes, a third of which were car accidents. As such, there is a clear need for cat owners to manage their cat's roaming behavior, for both environmental and cat welfare reasons.

Responsible cat ownership has been defined as "a commitment to perform various duties, sometimes specified by legislation, to satisfy a cat's behavioral, physical, and environmental needs while reducing risks that a cat may pose to the community, other animals, or the environment" [15 p. 2]. These duties are often considered to include cat containment [15], which reduces a cats' environmental impact and the likelihood of roaming-related accidents [15–17]. Cat containment is a cat management practice that includes a variety of behaviors and ranges from keeping a cat indoors at all times and providing controlled outdoor access (e.g., limiting a cat to an escape-proof fenced yard, an enclosure / run, or walks on a harness and lead), to keeping a cat indoors for a period of time (e.g., overnight) [4, 16]. Some authors, however, argue that the benefits of cat containment do not outweigh the negative impact it can have on a cat's wellbeing [18, 19] and that free roaming cats help to suppress rodent populations [20, 21]. Notwithstanding these criticisms, cat containment is endorsed by animal welfare organizations internationally as a means of protecting wildlife, improving a cat's welfare, preventing disease transmission and avoiding nuisance to neighbors [22–24].

Despite the support for cat containment, it a novel practice in NZ, with research finding that less than 14% of NZ owners kept their cat indoors or on their property at all times and only 14–29% inside overnight [16, 25]. Interestingly, the prevalence of cat containment has been found to vary significantly by country and region [25, 26]. According to Hall and colleagues [25], approximately two thirds of owners in NZ (67%) and the United Kingdom (UK; 64%) let their cat roam freely, while owners in Australia (80–92%) and the United States (80–93%) commonly performed some form of cat containment (indoors overnight, controlled outdoor access, or indoors at all times), and in Japan most kept their cat indoors (75%).

To improve cat management practices in NZ, evidence-based interventions may be necessary to change cat owner's behavior. Interestingly, McLeod et al., [27] who conducted an audit of the cat management interventions employed by organizations internationally, found that many effective persuasive communication techniques (e.g., framing, credible messenger, goal setting) were under-utilized. They found that most organizations relied on logical, evidenced-based messaging to educate cat owners. While education helps to increase knowledge and awareness of an issue, this alone is often unlikely to change behavior due to other barriers that may exist (i.e., habits, emotions, situational and contextual influences) [28–30]. This demonstrates the importance of utilizing behavioral theory to design effective behavioral interventions.

Community based social marketing (CBSM) is an approach that seeks to apply social and behavioral science to promote positive change in a community. In accordance with this framework, to effectively address animal management issues behavior change interventions should follow these four key steps: 1) select a human behavior to target which has the greatest potential to address the issue, 2) determine the barriers and drivers underlying non-participation in the prioritized behavior, 3) develop interventions that, according to behavioral theory, directly address the identified barriers and drivers to increase engagement in the behavior, and 4) evaluate the effectiveness of the intervention in promoting behavior change [29, 31, 32]. For the issue of free-roaming companion cats in NZ, Linklater et al., [16] conducted the first step by applying the McKenzie-Mohr behavior prioritization framework [33]. They considered the beneficial impact of various behaviors, their likelihood of adoption among owners, the proportion of owners already performing them, and veterinarians' opinions. Of the behaviors investigated they found that cat containment was the most effective behavior to target and promote among NZ cat owners (particularly keeping cats on the property always and overnight). With cat containment supported as the best behavior to target, the second step is to understand the barriers and drivers of cat containment, which can then help to design interventions to increase participation in this behavior in NZ [29, 31, 32, 34].

Behavioral theory can help us to understand what barriers and drivers influence cat owners' behavior and inform the design of interventions to increase uptake of cat containment. Selecting an appropriate framework, however, is no simple feat, with Michie and colleagues identifying 83 theories that seek to explain human behaviour [35]. McLeod and colleagues [36] reviewed nine behavioural theories relevant to invasive animal management (health belief model, protection motivation theory, theory of reasoned action and theory of planned behavior (TPB), focus theory of normative conduct, theory of interpersonal behavior, norm activation theory, value belief norm theory, affect heuristic, and needs, opportunities and abilities model). They noted that each of these models have their own limitations. For instance, the TPB [37], a general model of behavior that has previously been applied to cat containment [38, 39], does not consider other factors known to cause behavior, such as situational, habitual, and emotive factors [36]. Furthermore, many of these behavioral models do not directly link the causes of behavior with a means of addressing them. As such, McLeod and colleagues [36] proposed utilising an overarching framework, such as the COM-B model and Behaviour Change Wheel (BCW) [30] to firstly integrate the barriers and drivers of behaviour specified by these disparate theories and secondly link these behavioural determinants with interventions to address them.

A broad range of domains have applied the COM-B model [30] to understand the barriers and drivers of behavior related to health [40–46], pro-environmental behavior [47–49], agriculture [50, 51], and invasive animal management [52–54]. COM-B is an integrative model developed by Michie and colleagues [30], which builds on previous behavior change frameworks and states that behavior is the result of three main factors: capability, opportunity, and motivation (COM). Capability is a person's psychological and physical capacity to perform a behavior and it includes factors such as skills, awareness, knowledge, and confidence in one's ability to perform a behavior (i.e. perceived behavioral control [37], self-efficacy [55]). Opportunity is comprised of physical and social opportunity, which are external factors that enable a behavior to be performed. Physical opportunity includes having the necessary time, environmental and financial circumstances, and access to resources required to perform a behavior. Social Opportunity includes social norms and influences that make performance of a behavior more or less likely. Social norms can be further classified as descriptive (i.e., what is typically done by others) or injunctive (i.e., what others typically approve or disapprove of) [56]. Motivation includes internal factors that direct a behavior and is comprised of reflective motivation

(the conscious beliefs, attitudes, and goals which inform decision making) and automatic motivation (the emotional associations, habits, and impulses that subtly direct behavior).

An additional strength of the COM-B model over other behavioral theories, is that it demonstrates how behavior can be modified by linking known barriers and drivers with appropriate interventions to address them through the BCW [30, 36]. For instance, according to the BCW, if non-participation in a desired behavior is the result of low physical capability, techniques such as training or enablement are recommended to increase physical capability and thereby change behavior. This is consistent with CBSM which discusses that to change behavior, focus must be given to addressing the barriers and drivers associated with it [29]. For these reasons, the COM-B model has been applied in this study to understand the behavior of cat owners and identify leverage points for interventions.

Audience segmentation analysis can also be conducted to develop an in-depth understanding of the barriers and drivers of cat containment [31, 32]. Audience segmentation allows for the determination of whether a population is a single audience with the same set of factors impacting behavior, or whether multiple segments exist, each with a different set of factors influencing behavior [31, 32, 57]. While this is an approach commonly used by marketers, it has also been utilized in the context of climate change communications [58–60], pro-environmental behavior [61, 62], health [63], and invasive animal management [52–54, 64]. Previous research on invasive animal management has demonstrated that populations are not homogenous, with multiple segments identified that varied in their level of support for a desired behavior [53, 54, 64]. Identifying unique audiences can help to decide which segments to target for behavior change and whether interventions should be tailored to suit the characteristics of each audience [31, 32].

Preliminary research has sought to identify the barriers and drivers of owners' cat containment behavior and to determine whether there are distinct audiences of cat owners. In the NZ context, MacDonald et al., [38] applied the theory of planned behavior (TPB) to understand the influences on owners' intentions to keep their cat inside at night. They found that 31% of owners contained their cat at night and that the most influential predictors of night-containment intentions were attitudes, particularly the belief that night-containment is beneficial for the cat. They also found that household members and veterinarians influenced through injunctive norms night-containment intentions and that other cat owners' lack of engagement in night-containment predicted non-night-containment intentions. These findings suggest that attitudes and norms may be drivers of night-containment behavior.

International research offers additional insight into the influences on cat containment behavior and the types of audience segments that may exist in NZ. Firstly, Tan et al., [26] found that perceived benefits and risk of roaming were related to allowing cats to free roam. In addition, demographics and cat characteristics were related to cat containment behavior, which was also demonstrated by an international study [65]. In the Australian context, van Eeden et al., [39] applied the TPB and found that 83% of cat owners surveyed in the state of Victoria performed cat containment (30% indoors overnight, 53% controlled outdoor access or indoors at all times). The most important predictors of this behavior included perceived ability to perform cat containment and beliefs (concerns about cat safety, cats' right to roam, and cat predation behavior). In the UK, Crowley et al., [64] identified five distinct cat owner segments based on their beliefs, attitudes, and emotional reactions related to their cat's roaming and predation behavior. Their findings suggested that a key motivator to perform cat containment for some groups was cat welfare, while for others it was reducing predation of wildlife. Lastly, McLeod et al., [52] found that owners surveyed in Australia can be segmented into four distinct audiences based on their cat containment behavior, and that the majority contained their cat at all times or overnight (65%). Owners who *always* contained their cat

(33%) were more confident in their ability to perform containment, had more positive attitudes towards the benefit of cat containment, and were more likely to contain their cat if it was compulsory (injunctive social norm), compared to those who only had a *night curfew* for their cat (32%), *sporadically* (29%), or *never* (6%) contained their cat.

Overall, previous research [26, 38, 39, 52, 64] supports that COM factors such as psychological capability, social norms, beliefs, attitudes, and emotional reactions predict cat containment behavior, and that distinct audiences of cat owners exist which differ in COM factors. However, these studies have several limitations which must be noted. Firstly, in NZ MacDonald et al., [38] only investigated night containment intentions, and so the influences on NZ cat owners' performance of the full scope of cat containment behaviors remain unknown. Secondly, research has primarily focused on assessing the impact of motivation and social norms on cat containment [26, 38, 39, 64], with only one study having assessed a range of COM factors [52]. Thirdly, aside from one study [26], previous study populations were limited to either one [38, 39, 52] or two localities [64], meaning previous findings may not be generalizable to wider populations of cat owners. Finally, only one study [52] assessed whether distinct segments of cat owners differed in their behavior. However, this study [52] mainly highlighted differences between participants and non-participants in cat containment, likely due to by the high participation in cat containment (65%) that was found [52]. As such, how non-participants in cat containment may differ from one another remains unclear.

While preliminary research internationally has assessed the barriers and drivers of cat containment and conducted audience segmentation analysis, further research is still required in the NZ context for several reasons. Firstly, the majority of NZ owners do not participate in cat containment [16, 25], and so it is not yet known how COM-B profiles differ in a large population of non-participants. Secondly, Hall et al., [25] in an international survey of cat owners found that attitudes towards various cat management practices and perception of the issues posed by companion cats to wildlife varied significantly by country. In Australia, for instance, the majority of owners believed there is a need for cat management legislation and that cats should be kept inside at night, compared with less than 40% of NZ owners and only around 20% of UK owners. Furthermore, nearly two-thirds of Australian owners perceived companion cats killing wildlife to be a serious issue, compared to less than half of NZ owners and only around 10% of UK owners. According to the COM-B model [30] these differing beliefs suggest that NZ owners may be less motivated to contain their cats than AU owners, but more so than UK owners. Therefore, we can conclude that despite the cultural and environmental similarities between NZ, Australia, and the UK, there are likely different factors influencing the adoption of cat containment behavior and different segments of cat owners in each country.

In summary, previous research has identified barriers and drivers that influence owners' adoption of cat containment and that distinct audiences of owners may exist with unique COM-B profiles [26, 38, 39, 52, 64]. However, despite this body of research, two key questions remain unanswered. Firstly, what is the comprehensive set of COM factors that act as barriers and drivers of NZ owners' participation in cat containment, and secondly can NZ owners be segmented into unique audiences based on their COM profiles, that differ significantly in their performance of cat containment. This study sought to address these gaps in the literature using data from a quantitative cross-sectional online survey of NZ cat owners [66]. Consistent with the COM-B model of behavior [30] and previous research in invasive animal management [26, 31, 36, 38, 39, 52–54, 64], the following hypotheses were made: 1) cat owners with increased capability, opportunity, and motivation to contain will have greater cat containment intentions and behavior, and 2) at least three segments of cat owners will exist in NZ which differ significantly in the set of COM factors predicting their cat containment behavior. The findings from this study will inform the design of empirically grounded behavioral interventions

that address the causes of non-participation in cat containment for different audiences of cat owners in NZ, which will thereby improve the management of companion cats.

# Method

## Participants

Participants were recruited using a Lucid Marketplace online sample [67] (between (February 3, 2022 and February 10, 2022) and comprised 395 cat owners residing in NZ, aged 18 years and over. Reviewing previous research that assessed the relationship between barriers and drivers of cat containment and behavior and intentions demonstrated that the smallest expected effect size for this study was $f^2$ = 0.20 [38, 68, 69]. Furthermore, a previous study that segmented cat owners into audience groups based on their cat containment beliefs found a five segment solution [64]. As such a power analysis using G*Power [70], with a power level of 0.90 and an α level of 0.05, was conducted based on an expected five segment solution. The power analysis indicated at least 390 participants were required for adequate statistical power.

An online panel sample was selected as it offered the potential to recruit the large number of participants required within the time constraints of this project and circumvent issues with COVID-19 lockdowns (stay at home orders) occurring at this time in NZ. Notwithstanding, there are known limitations with online panel samples pertaining to data quality and external validity [71]. Various measures were implemented in the study design to ensure participant data quality, including screening questions to confirm current cat ownership and age, screening out those that completed the survey in less than four minutes or failed any of three attention check questions, as well as various security checks (bot and duplicate detection). External validity is another concern for online panel samples due to lack of sample representativeness [71] and is a limitation of this research. However, meta-analyses have supported that online panel samples have equivalent external validity to other forms of convenience samples traditionally used in applied psychology research [71].

In the sample 254 participants identified as female (64%), 137 as male (35%) and four as other (1%) [66]. The mean year of birth was 1974 (*SD* = 17.18, range = 1931 to 2004), meaning participants were approximately 48 years old on average (*Median* = 48). Most participants identified as NZ European (76%), followed by Māori (10%), and other (14%). The majority had an undergraduate qualification or greater (38%), followed by a secondary school qualification (29%). Compared to the 2018 NZ census, females and NZ European were over-represented, and participants had an older median age and were more educated (NZ 2018 census: 51% female; median age of 37; 25% with an undergraduate qualification or greater; 70% NZ European [72]). However, according to Companion Animals New Zealand [1], this is consistent with NZ cat owners. They found that cat ownership rates were highest in females (43%) and adults aged 35–64 years (43%-53%) compared with New Zealanders overall (41%). Moreover, cat ownership was significantly higher among NZ Europeans (46%) compared to other ethnic groups such as Māori (36%).

**Procedure and materials.** An online survey was conducted in February 2022 using the Qualtrics online survey platform [73]. The study received blanket ethics approval from the Human Ethics Committee of the University of Canterbury. Participants' informed written consent was obtained at the beginning of the questionnaire. Demographics information was collected first, then definitions of 'free-roaming cats', 'cat containment', 'cat enclosure', and 'cat escape-proof fence' were provided, and after that cat containment intentions and behaviors were assessed. Finally, items pertaining to COM to perform cat containment were completed. Participants who did not pass the security checks (e.g., bot and duplicate response detection; *n* = 23), did not currently own a cat (*n* = 508), were born after 2004 (*n* = 1), and

completed the questionnaire in less than 4 minutes or failed the three attention check items (e.g., "to show that you are paying attention, we ask you to select 'agree'"; $n = 112$) were screened out. The questionnaire is shown in S1 Appendix.

## Measures

**Cat containment intentions and behavior.** Cat containment was assessed with two subscales, Containment Intentions and Containment Behavior. Containment Intentions were measured with seven items asking participants how often in the next six months they expected to do the following behaviors: allow cat to roam freely, keep cat indoors, keep cat indoors overnight, confine cat to an escape-proof fenced yard when outside, confine cat to an enclosure when outside, walk cat on a harness and lead when outside, and fully supervise cat when outside. After reverse scoring the free-roaming item, the items were averaged into a single scale which demonstrated adequate internal consistency (Cronbach's $\alpha = 0.84$). Containment Behavior was assessed with seven items asking participants how often they currently did the following behaviors: allow cat to roam freely, keep cat indoors, keep cat indoors overnight, confine cat to an escape-proof fenced yard when outside, confine cat to an enclosure when outside, walk cat on a harness and lead when outside, and fully supervise cat when outside [52]. After reverse scoring the free-roaming item, the items were averaged into a single scale, which demonstrated adequate internal consistency (Cronbach's $\alpha = 0.83$). All responses were assessed on a five-point Likert scale (1 = never, 2 = rarely, 3 = sometimes, 4 = often, 5 = always).

**COM variables.** To assess cat owners' capability, opportunity, and motivation related to containment, seven subscales were developed with 51 items (S1 Table). The items were drawn or adapted from previous qualitative and quantitative research on barriers and drivers of cat containment [14, 52, 64, 68, 69]. To ensure all subdimensions of COM were comprehensively assessed, additional items were also drawn from related invasive species management research or developed based on behavioral theory [30, 32, 36, 54].

Capability was assessed with a single subscale, Capability to Contain, which comprised nine items measuring cat owners' physical capability, awareness and knowledge of the issue, memory / attention capacity, and behavioral regulation towards cat containment. All responses were assessed on a five-point Likert scale (1 = strongly disagree, 2 = disagree, 3 = neither agree nor disagree, 4 = agree, 5 = strongly agree). Five items were reverse scored so that higher scores on all items reflected greater capability to contain. Following reverse scoring, the items were averaged to create a single scale, which demonstrated adequate internal consistency (Cronbach's $\alpha = 0.70$).

Opportunity was assessed with two subscales, Physical and Social Opportunity to Contain. Physical Opportunity to Contain was measured with eight items that assessed time, resource availability, and environmental context relevant to cat containment. These responses were assessed on a five-point Likert scale (1 = strongly agree, 2 = agree, 3 = neither agree nor disagree, 4 = disagree, 5 = strongly disagree), with higher scores reflecting greater physical opportunity to contain. The scores for the items were averaged to create a single scale, which demonstrated adequate internal consistency (Cronbach's $\alpha = 0.81$). Social Opportunity to Contain was measured with four items. The items assessed interpersonal influences on cat containment (e.g., veterinarians, other cat owners). Responses were assessed on a five-point Likert scale (1 = strongly oppose, 2 = oppose, 3 = neutral, 4 = support, 5 = strongly support), with higher scores indicating greater social opportunity to contain. The scores for the items were averaged to create a single scale, which demonstrated adequate internal consistency (Cronbach's $\alpha = 0.86$).

Motivation was assessed with four subscales; Concern About Roaming, Containment is Beneficial for Cat Beliefs, Pro-Containment Beliefs, and Automatic Motivation to Contain.

Concern About Roaming was measured using 11 items, assessing perceived levels of concern about potential issues related to cats roaming freely, such as injury or death on the road and killing wildlife. Responses were assessed on a five-point Likert scale (1 = unconcerned, 2 = slightly concerned, 3 = somewhat concerned, 4 = very concerned, 5 = extremely concerned), with higher scores indicating greater concern about consequences of roaming. The scores for the items were then averaged to create a single scale which demonstrated high internal consistency (Cronbach's α = 0.95). Containment is Beneficial for Cat Beliefs was measured with four items, assessing the perceived impact of containment on a cat's quality of life. Responses were assessed on a five-point Likert scale (1 = very harmful, 2 = harmful, 3 = neither beneficial nor beneficial, 4 = beneficial, 5 = very beneficial) with higher scores indicating greater perceived benefit of containment for cats. The scores for the items were averaged to create a single scale which demonstrated adequate internal consistency (Cronbach's α = 0.80). Pro-Containment Beliefs were measured with 13 items, assessing beliefs and attitudes towards containment and roaming, perceived effort of containment, and social role identity. All responses were assessed on a five-point Likert scale (1 = strongly disagree, 2 = disagree, 3 = neither agree nor disagree, 4 = agree, 5 = strongly agree). Nine items were reverse scored so that higher scores on all items represented stronger pro-containment beliefs. The scores for the items were averaged to create a single scale which demonstrated high internal consistency (Cronbach's α = 0.90). Automatic Motivation to Contain was measured with two items, assessing emotions towards containment and containment habits on a five-point Likert scale (1 = strongly agree, 2 = agree, 3 = neither agree nor disagree, 4 = disagree, 5 = strongly disagree), with higher scores indicating stronger automatic motivation to contain. The scores for the items were averaged to create a single scale, which demonstrated adequate internal consistency (Cronbach's α = 0.81).

## Statistical analyses

Descriptive analysis and bivariate correlations were conducted to assess relationships among all variables, followed by multiple regression analyses to determine the degree to which COM variables predicted Containment Intentions and Behavior. Latent profile analysis (LPA) was then used to classify participants into homogenous subgroups based on their scores on COM variables. Akaike's Information Criterion (AIC) and Entropy were used to assess the relative model fit of the profile solutions and the Lo-Mendell-Rubin (LMR) test was used for model comparison. Lower AIC values indicate better model fit whilst lower Entropy values indicate greater uncertainty in the model [74–77]. LMR is a significance test that compares the likelihood ratio of a model with one which has less profile groups [77, 78]. AIC, Entropy and LMR were all evaluated to select a profile solution to retain for interpretation. Multivariate analysis of variance (MANOVA) was then conducted to validate the profiles and determine what proportion of variance in the cat containment behavioral variables was explained by the subgroups. Follow up analyses of variance (ANOVAs) using the Welche's $F$ test were then used to test for significant differences in the two cat containment behavioral variables between the segments. To determine which groups differed significantly, post hoc comparisons using the Games-Howell test was conducted. Mplus 7.0 was used to conduct the LPA [79] and IBM SPSS Statistics 26.0 was used for all other analyses [80]. For all significance testing a critical alpha value of 0.05 was used.

## Results

### Descriptive statistics & bivariate correlations for study variables

All COM variables were found to have moderate to large significant associations with Containment Intentions and Behavior ($r > 0.30$; Table 1) [81]. As expected, increases in levels of the

**Table 1. Descriptive statistics and correlations for cat owners' intentions and behavior and capability, opportunity, and motivation to perform cat containment.**

| Variable | M | SD | 1 | 2 | 3 | 4 | 5 | 6 | 7 | 8 | 9 |
|---|---|---|---|---|---|---|---|---|---|---|---|
| 1. Cat Containment Behavior | 1.85 | 0.75 | — | | | | | | | | |
| 2. Cat Containment Intentions | 1.87 | 0.80 | 0.95* [0.92, 0.96] | — | | | | | | | |
| 3. Capability to Contain Cat | 3.15 | 0.65 | 0.49* [0.41, 0.56] | 0.50* [0.42, 0.57] | — | | | | | | |
| 4. Physical Opportunity to Contain Cat | 2.87 | 0.92 | 0.33* [0.24, 0.42] | 0.34* [0.26, 0.42] | 0.61* [0.53, 0.67] | — | | | | | |
| 5. Social Opportunity to Contain Cat | 2.69 | 0.85 | 0.48* [0.39, 0.56] | 0.50* [0.42, 0.57] | 0.30* [0.19, 0.39] | 0.22* [0.12, 0.32] | — | | | | |
| 6. Concern About Cat Roaming Beliefs | 2.78 | 1.06 | 0.49* [0.41, 0.57] | 0.50* [0.42, 0.58] | 0.30* [0.20, 0.40] | 0.18* [0.07, 0.28] | 0.38* [0.28, 0.47] | — | | | |
| 7. Containment is Beneficial for Cat Beliefs | 2.59 | 0.79 | 0.66* [0.60, 0.71] | 0.66* [0.60, 0.71] | 0.45* [0.36, 0.53] | 0.32* [0.22, 0.41] | 0.61* [0.54, 0.67] | 0.48* [0.39, 0.56] | — | | |
| 8. Pro Cat Containment Beliefs | 2.59 | 0.72 | 0.66* [0.60, 0.73] | 0.68* [0.62, 0.74] | 0.60* [0.51, 0.66] | 0.44* [0.35, 0.53] | 0.64* [0.57, 0.71] | 0.49* [0.40, 0.58] | 0.73* [0.68, 0.77] | — | |
| 9. Automatic Motivation to Contain Cat | 2.07 | 1.03 | 0.63* [0.55, 0.70] | 0.64* [0.57, 0.70] | 0.55* [0.46, 0.62] | 0.41* [0.32, 0.50] | 0.53* [0.43, 0.61] | 0.32* [0.22, 0.41] | 0.63* [0.55, 0.70] | 0.80* [0.74, 0.84] | — |

$N = 395$. Containment / contain refers behaviors that cat owners can undertake to ensure their cat(s) remain on their property. Roaming refers to allowing cats to leave an owner's property. $M$ = mean. $SD$ = standard deviation. Pearson's correlation coefficient, $r = 0.30$ specifies a medium effect size and $r = 0.50$ a large effect size [81]. Values in square brackets indicate bias corrected and accelerated bootstrap 95% confidence interval per correlation. All variables range from a minimum 1 to maximum 5.

* $p < 0.05$ (2-tailed).

COM variables were all associated with increases in cat owners' cat containment intentions and behavior (Table 1).

## Predicting cat containment behavior with the COM variables

Multiple regression was used to investigate the extent to which COM factors predicted cat containment intentions and behavior. In the first analysis, the factors Capability to Contain, Physical Opportunity to Contain, Social Opportunity to Contain, Concern About Roaming, Containment is Beneficial for Cat Beliefs, Pro-Containment Beliefs, and Automatic Motivation to Contain were used to predict Containment Intentions, and in the second analysis these factors were used to predict Containment Behavior. All variables were simultaneously entered into the models. All predictors had Variance Inflation Factors below 5 and Tolerance above 0.20 indicating that no variables needed to dropped from the regression models to reduce multicollinearity [82, 83].

The results from the multiple regression analyses found that Capability to Contain, Concern About Roaming, Containment is Beneficial for Cat Beliefs, and Automatic Motivation to Contain were all significant predictors of Containment Intentions and Containment Behavior (Tables 2 and 3). These results demonstrate that cat owners intended to perform and performed cat containment to a greater extent when they: 1) had the psychological and physical capability to contain their cat, 2) were concerned about the negative consequences of roaming, 3) believed containment is beneficial for a cat, and 4) had automatic motivation to perform cat containment. While Physical Opportunity to Contain, Social Opportunity to Contain, and Pro-Containment Beliefs were not significant predictors of Containment Intentions and Behavior, they were all individually significantly associated with Containment Intentions and Behavior (Table 1). Overall, the models explained 57% of the variance in Containment

Intentions and 56% of the variance in Containment Behavior. Of the variance explained in Containment Intentions, Containment is Beneficial for Cat Beliefs and Concern About Roaming each uniquely explained 3%, Automatic Motivation to Contain 2%, and Capability to Contain 1%. Of the variance explained in Containment Behavior, Containment is Beneficial for Cat Beliefs uniquely explained 4%, Concern About Roaming 3%, Automatic Motivation to Contain 2%, and Capability to Contain 1%.

## Segmentation of cat owners based on COM variables

LPA was conducted to determine whether there are multiple unique segments of cat owners, that differ in their COM to perform cat containment. The results identified four different cat owner profiles based on the COM variables. Using LPA, model fit indices pertaining to two to five profiles solutions were compared (Table 4). The five-profile solution had the lowest AIC values, indicating it had the best model fit, however, LMR indicated that a five-profile solution did not significantly better fit the data than a four-profile solution. In addition, with an Entropy value greater than 0.80, the four-profile solution is supported to have minimal uncertainty in its profile classification of individuals (according to Celeux and Soromenho, and Tein et al., as cited in Ferguson et al., [77]). Therefore, the four-profile solution was retained for interpretation. The segments were labelled; *engaged*, *receptive*, *ambivalent*, and *opposed*.

The smallest segment *engaged* ($n$ = 22, 6%), had the highest levels of all COM variables relative to the other segments (Fig 1). Members of this segment had the strongest Automatic Motivation to Contain, Pro-Containment Beliefs, and Capability to Contain. The *receptive* segment ($n$ = 68, 17%) had the second highest levels of all COM variables. They had moderate Automatic Motivation to Contain, Pro-Containment Beliefs, Containment is Beneficial for Cat Beliefs, and Social Opportunity to Contain. Their Capability to Contain and Physical Opportunity to Contain, while still above average, were however relatively lower than the other COM factors. The largest segment, *ambivalent* ($n$ = 188, 48%), had overall average levels of the COM variables. Their Social Opportunity to Contain, Concern About Roaming, and Containment is Beneficial for Cat Beliefs were slightly above average relative to others in the sample. In addition, their Capability to Contain, Physical Opportunity to Contain, and Automatic Motivation

**Table 2. Predicting cat containment intentions from capability, opportunity and motivation factors of containment.**

| Predictors | B | 95% CI for B | | SE B | $\beta$ | $sr^2$ |
|---|---|---|---|---|---|---|
| | | LB | UB | | | |
| Constant | -0.47 | -0.77 | -0.17 | 0.15 | - | - |
| Capability to Contain | 0.13* | 0.01 | 0.25 | 0.06 | -0.11 | 0.01 |
| Physical Opportunity to Contain | -0.00 | -0.07 | 0.07 | 0.04 | -0.00 | 0.00 |
| Social Opportunity to Contain | 0.03 | -0.05 | 0.11 | 0.04 | 0.03 | 0.00 |
| Concern About Roaming | 0.14* | 0.09 | 0.20 | 0.30 | 0.20 | 0.03 |
| Containment is Beneficial for Cat Beliefs | 0.27* | 0.16 | 0.37 | 0.05 | 0.26 | 0.03 |
| Pro-Containment Beliefs | 0.15 | -0.01 | 0.30 | 0.08 | 0.13 | 0.00 |
| Automatic Motivation to Contain | 0.18* | 0.10 | 0.27 | 0.04 | 0.24 | 0.02 |

$N$ = 395. Containment / contain refers to behaviors that cat owners can undertake to ensure their cat(s) remain on their property. Roaming refers to allowing cats to leave an owner's property. B = unstandardized beta coefficients. CI = confidence interval. LB = lower bound. UB = upper bound. $\beta$ = standardized beta coefficients. $sr^2$ = squared semi-partial correlation coefficient (the proportion of variance the predictor uniquely explained in the dependent variable over and above the other predictors). All predictors ranged from a minimum 1 to maximum 5.

$R^2$ = 0.57. Adjusted $R^2$ = 0.57.

* $p < 0.05$

**Table 3. Predicting containment behavior from capability, opportunity and motivation factors of cat containment.**

| Predictors | B | 95% CI for B | | SE B | $\beta$ | $sr^2$ |
|---|---|---|---|---|---|---|
| | | LB | UB | | | |
| Constant | -0.29 | -0.57 | -0.00 | 0.15 | - | - |
| Capability to Contain | 0.13* | 0.02 | 0.24 | 0.06 | -0.01 | 0.01 |
| Physical Opportunity to Contain | -0.01 | -0.08 | 0.06 | 0.04 | -0.01 | 0.00 |
| Social Opportunity to Contain | 0.01 | -0.08 | 0.09 | 0.04 | 0.01 | 0.00 |
| Concern About Roaming | 0.14* | 0.08 | 0.19 | 0.30 | 0.20 | 0.03 |
| Containment is Beneficial for Cat Beliefs | 0.29* | 0.19 | 0.39 | 0.05 | 0.30 | 0.04 |
| Pro-Containment Beliefs | 0.09 | -0.06 | 0.24 | 0.08 | 0.08 | 0.00 |
| Automatic Motivation to Contain | 0.18* | 0.10 | 0.27 | 0.04 | 0.25 | 0.02 |

$N$ = 395. Containment / contain refers to behaviors that cat owners can undertake to ensure their cat(s) remain on their property. Roaming refers to allowing cats to leave an owner's property. B = unstandardized beta coefficients. CI = confidence interval. LB = lower bound. UB = upper bound. $\beta$ = standardized beta coefficients. $sr^2$ = squared semi-partial correlation coefficient (the proportion of variance the predictor uniquely explained in the dependent variable over and above the other predictors). All predictors ranged from a minimum 1 to maximum 5.

$R^2$ = 0.56. Adjusted $R^2$ = 0.55.

* $p < 0.05$

to Contain were slightly below average. The *opposed* segment ($n$ = 117, 30%) had the lowest overall levels of the COM variables. They had the lowest Pro-Containment Beliefs, Containment is Beneficial for Cat Beliefs, and Social Opportunity to Contain.

**Demographic characteristics of cat owner profiles.** In terms of demographic characteristics (S2 Table), one-way ANOVAs found that, on average, members of the segments did not differ significantly in their year of birth, $F(3, 391)$ = 1.98, $p$ = 0.116, education level, $F(3, 391)$ = 1.56, $p$ = 0.199) or number of cats owned $F(3, 391)$ = 1.24, $p$ = 0.296. We did not conduct non-parametric statistical analyses on the remaining demographic variables (locality, type of dwelling, gender, home ownership) given that, except in one case, the group sizes were too small. That is, the minimum expected count was not greater than 1 and more than 20% of expected counts were less than 5% [84, 85].

## Validation of cat owner segments

The relationship between membership in the four cat owner segments and the cat containment behavioral variables were then examined using MANOVA to confirm if segment membership

**Table 4. Model fit indices for two to five cat owner profile solutions.**

| Profile Solution | AIC | Δ AIC | Entropy | LMR | $p$ |
|---|---|---|---|---|---|
| 1 | 6942.20 | - | - | - | - |
| 2 | 6171.47 | 770.73 | 0.91 | 770.62 | 0.070 |
| 3 | 5823.78 | 347.69 | 0.85 | 356.24 | 0.380 |
| 4 | 5659.76 | 164.02 | 0.87 | 176.33 | 0.008 |
| 5 | 5621.56 | 38.20 | 0.84 | 53.15 | 0.550 |

AIC = Akaike's Information Criterion. Δ AIC = the difference between the AIC for a solution $k$ profiles retained with the AIC associated with the solution $k$-1 profiles retained. LMR = Lo-Mendel-Rubin likelihood ratio test. Lower values of AIC indicate better model fit, and low entropy values indicate more uncertainty in a model's classification of individuals [74–77]. LMR is a significance test that compares the likelihood ratio of a model with one that has less profile groups [78].

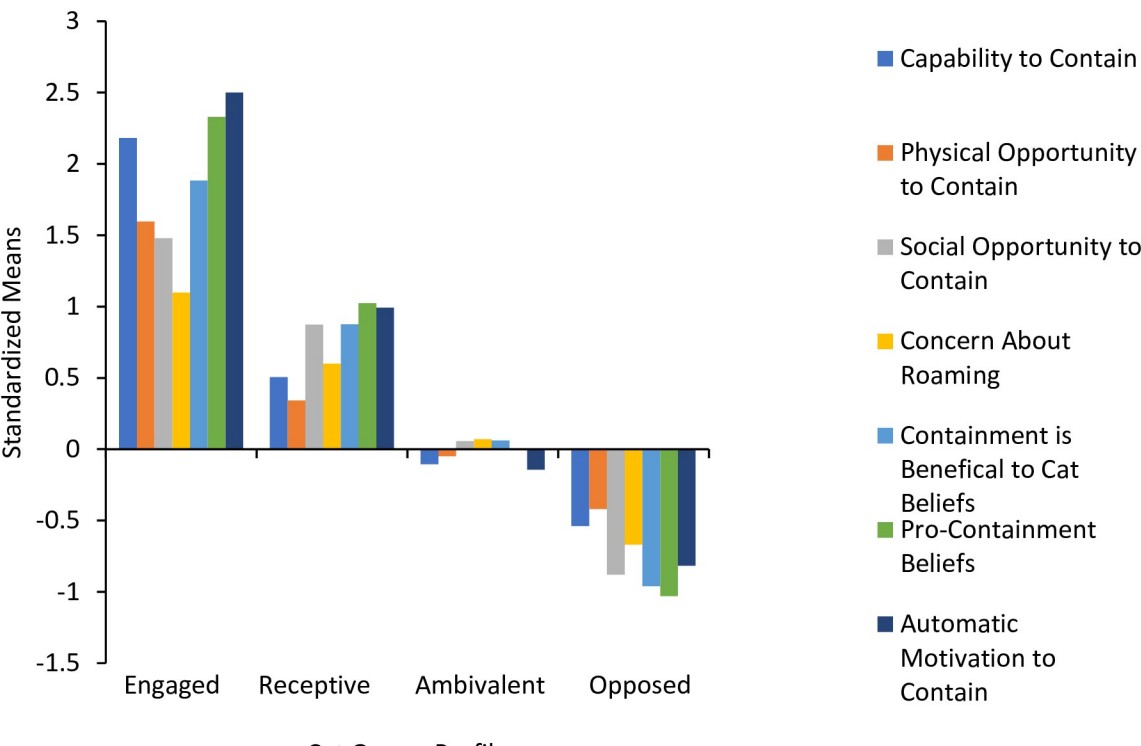

**Fig 1. Standardized means of the capability, opportunity, and motivation factors of cat containment across the four cat owner profiles.** *Notes*: Engaged, n = 22. Receptive, n = 68. Ambivalent, n = 188. Opposed, n = 117. Containment / contain refers to a variety of behaviors that cat owners can undertake to ensure their cat(s) remain on their property. Roaming refers to allowing cats to leave an owner's property. Standardized means reflect the segment's deviation from the sample mean on the variables.

predicted cat containment. Two follow up ANOVAs with post-hoc tests were then used to test for significant differences in Containment Intentions and Behavior between the segments.

The MANOVA demonstrated that the four cat owner profiles explained 31% of the variance in the cat containment behavioral variables, $\eta^2 = 0.31$, $V = 0.47$, $F(6, 780) = 59.13$, $p < 0.001$. Two follow up one-way ANOVAs were then conducted to determine the group effect for each behavioral variable. Levene's test for equality of variance was significant for Containment Intentions ($p < 0.001$) and Containment Behavior ($p < 0.001$). Given the heterogeneity of variances, the more robust Welch's $F$ test was used for the univariate significance tests [85]. A significant effect of group on levels of Containment Intentions, $\eta^2 = 0.52$, $F(3, 79.55) = 107.28$, $p < 0.001$, and on levels of Containment Behavior $\eta^2 = 0.49$, $F(3, 80.05) = 89.07$, $p < 0.001$ was found.

To determine which groups differed significantly, post-hoc comparisons using the Games-Howell test, which is considered accurate when population variances are unequal, were conducted (according to Toothaker, as cited in Field [85]). All segments were found to differ significantly from each other in their Containment Intentions ($p \leq 0.005$) and Containment Behavior ($p \leq 0.003$). As shown in Fig 2, the *engaged* segment had the highest levels of both Containment Intentions ($M = 3.35$, $SD = 0.71$) and Containment Behavior ($M = 3.27$, $SD = 0.73$), followed by *receptive* (Containment Intentions $M = 2.70$, $SD = 0.87$; Containment Behavior $M = 2.58$, $SD = 0.77$), *ambivalent* (Containment Intentions $M = 1.73$, $SD = 0.50$; Containment Behavior $M = 1.72$; $SD = 0.50$), and *opposed* (Containment Intentions $M = 1.34$,

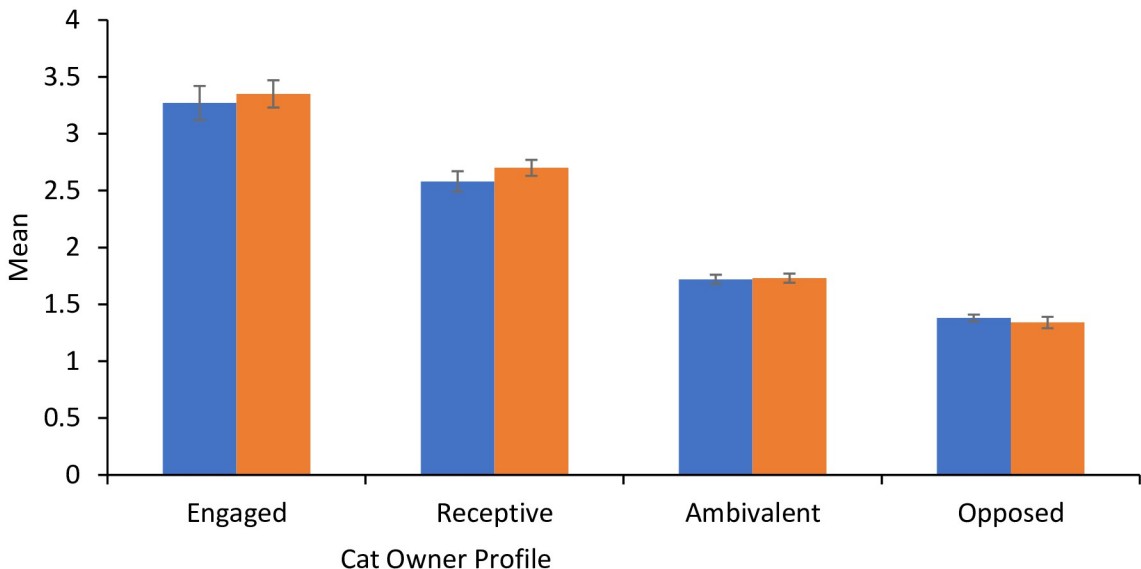

**Fig 2. Means of the cat containment intentions and behavior variables for the four cat owner profiles.** Notes: Containment behavior is shown in blue and Containment Intentions in orange. Engaged, $n = 22$. Receptive, $n = 68$. Ambivalent, $n = 188$. Opposed, $n = 117$. The error bars represent 95% confidence intervals. All variables ranged from a minimum 1 to maximum 5.

$SD = 0.33$; Containment Behavior $M = 1.38$; $SD = 0.38$). These results support that distinct audiences of cat owners exist among surveyed owners with their cat containment intentions and behavior being determined by different COM profiles.

## Discussion

This study sought to identify the barriers and drivers of cat containment for NZ cat owners. We found that cat containment was rarely intended to be performed, and rarely performed by NZ cat owners. Furthermore, cat owners with greater capability, opportunity, and motivation (COM) to perform cat containment were more likely to contain their cat. Four audiences of cat owners who differed in the COM factors predicting their cat containment intentions and behavior were also identified. These findings are explored below, alongside a discussion of how practitioners can best target these segments to increase participation in cat containment.

Firstly, as hypothesized, all COM factors assessed in this study were positively associated with cat containment intentions and behavior. The factors that uniquely predicted intentions and behavior were Capability to Contain, Concern About Roaming, Containment is Beneficial for Cat Beliefs, and Automatic Motivation to Contain. As such, these findings demonstrate that cat containment was performed to a significantly greater extent when cat owners had the capability to perform containment (e.g., skills, memory, knowledge, awareness, and behavioral regulation), had physical and social opportunity enabling containment (e.g., time, environmental circumstances, and interpersonal influences), were concerned about the negative consequences of roaming (e.g., cat being injured or killing wildlife), believed that containment has a beneficial impact on a cat's quality of life, had pro-containment beliefs, and had greater automatic motivation to contain. These findings extends the literature by undertaking to the best of our knowledge, the most comprehensive assessment of cat owners' COM to perform cat containment to date and demonstrating that a range of additional factors are statistically

reliable predictors of cat containment intentions and behavior. This is consistent with CBSM [29, 34] and the COM-B model of behavior [30]. Furthermore, these results build upon previous findings which demonstrated that psychological capability, beliefs, attitudes, and social influences were predictors of cat containment [26, 38, 39, 52, 64, 69].

Secondly, as hypothesized, audience segmentation analysis demonstrated that NZ cat owners could be segmented into four subgroups, which differed significantly in the set of COM factors predicting their behavior. Consistent with the BCW [30], CBSM [29] and an audience segmentation approach [57], given these four segments have different sets of barriers and drivers to performing cat containment, tailored interventions should be designed to address the underlying causes of non-participation for each of these audiences [29, 31, 32, 34]. Research supports that tailored messages are more relevant and accessible to an audience, thus making behavior change more likely [86–89]. Furthermore, a theory-based approach to behavior change entails selecting an intervention that directly addresses the behavioral determinants that act as barriers and driver of the behavior [90]. Through applying COM-B and the BCW [30], it is possible to identify which interventions would be most suitable for reducing the barriers related to capability, opportunity and increasing motivation for each segment. The four cat owner segments were labelled; *engaged*, *receptive*, *ambivalent*, and *opposed*. *Engaged* cat owners (6%) were characterized by the highest COM factors and the highest participation in cat containment, followed by *receptive* (17%), and *ambivalent* cat owners (48%). *Opposed* cat owners (30%) were characterized by the lowest COM factors and the lowest participation in cat containment. Suitable behavior change techniques for each segment will be explored under the practical implications section that follows.

These findings have some similarities with other research, but also some key differences. Crowley et al., [64] identified five cat owner segments based on their beliefs, attitudes, and emotional reactions, however, did not consider situational and contextual factors. Furthermore, they did not evidence whether membership in these segments predicted behavior and the use of Q-methodology and a low sample size (*n* = 56*)* limits the generalizability of this study's findings. While McLeod et al., [52] also identified four cat-owner segments that differed significantly in their behavior, the largest segment identified were those who *always* contained their cat, while in the current study those who *engaged* in cat containment were the smallest segment. This suggests that NZ cat owners are mostly non-participants in cat containment compared to those in Australia, which is consistent with previous research [25]. NZ owners therefore appear to face greater barriers and less drivers of cat containment. In addition, McLeod et al., [52] found that the main COM differences were between the segment who *always* contained their cat and the segments that did not perform containment (*night curfew*, *sporadic*, and *never*). Those who *always* contained had stronger perceived behavioral control, beliefs about the benefit of containment to cats, and social normative influences than the other groups. The current study, however, found that all identified segments differed from each other on a broad set of COM factors. These differences may be due to the more comprehensive set of COM factors assessed in this study, leading to a more nuanced understanding of cat owners surveyed. Additionally, these findings may suggest that cross-cultural variations exist, with NZ cat owners, who are mainly non-participants in cat containment, being more heterogeneous on a larger range of COM factors than Australian cat owners. Reasons for this may include the success of cat containment campaigns that have taken place in Australia since the 1990s [16, 39, 52], the additional wildlife risks to cats in Australia (e.g., venomous snakes; [14]), and the more extensive cat management regulations in Australia compared to NZ [91].

## Practical implications

The findings from this study have several implications for practitioners seeking to increase participation in cat containment, such as government agencies and wildlife protection and animal welfare organizations. The identification of different audiences of cat owners suggests that it will be necessary to tailor interventions to each segment to effectively promote behavior change. Furthermore, in situations where resources are limited, a cost-effective approach could be targeting *receptive* and *ambivalent* cat owners with tailored interventions. Targeting these two segments could provide the best opportunity to exponentially increase cat containment practices in NZ given that they represented around two thirds of those surveyed and exhibited fewer barriers to change [29, 34].

The behavior change wheel (BCW) [30] can be applied to identify appropriate behavior change techniques for each segment [31, 32]. The BCW is a framework which links capability, opportunity, and motivational factors with strategies to address them [30]. For those *receptive* to cat containment, factors that need to be addressed were Capability to Contain, Physical Opportunity to Contain, and Concern About Roaming. According to the BCW [30], Capability to Contain could be increased through *training*, for instance, by offering workshops for cat owners on how to build their own cat enclosure. Physical Opportunity to Contain could be increased through *environmental restructuring* (e.g., improving the availability of cat containment products such as cat escape-proof fence systems). To increase Concern About Roaming, *education* could be used to inform cat owners about the prevalence of roaming-related accidents to cats [14]. The same techniques could also be applied to *ambivalent* cat owners, who also have low Capability to Contain, Physical Opportunity to Contain, and Concern About Roaming. However, consideration should also be given to improving their low Automatic Motivation to Contain, for instance through *modeling*. Influencers who perform cat containment in social media campaigns could act as role models to create positive associations with cat containment for *ambivalent* cat owners (e.g., showing off their 'catio' to their followers).

Whilst practitioners may prioritize targeting *receptive* and *ambivalent* cat owners, some consideration should also be given to *engaged* and *opposed* cat owners. *Opposed* cat owners could be shifted to become more *ambivalent* or even *receptive* to cat containment by using *persuasion* to address their lack of Pro-Containment Beliefs [27, 30]. This could involve using credible messengers, such as veterinarians, to communicate the benefits of enrichment for contained cats (e.g., providing food puzzles) and to persuade them that contained cats can be healthy and happy [30, 32, 92]. Finally, the techniques outlined could also be used to maintain the high participation in cat containment in *engaged* cat owners.

There are practical challenges of identifying and targeting members from each segment, which would need to be explored further at a local context. However, these segments could be reached using targeted advertising on social media or by focusing on consumer groups that map onto the four segments. For instance, groups opposed to cat management and / or which advocate for the need for cats to roam, or groups of consumers that visit websites providing advice or products pertaining to creating stimulating environments for contained cats. To select appropriate communication channels consideration should be given both to how well a channel can reach the targeted audience and whether the channel suits the characteristics of the message being delivered [93]. Finally, the ethical considerations of seeking to change cat owner behavior must also be considered, and it is recommended that an inclusive approach is taken, engaging with stakeholders to ensure there is community support for increasing the prevalence of cat containment and a behavior change program [31, 94].

Limitations of this research include the use of self-report measures of behavior and a cross-sectional study design. While self-report measures have a number of benefits, they can have a recall and social desirability bias, as has been found in the assessment of lifestyle behaviors [95, 96], and thus lead to over- or under-reporting of behavior. In addition, the cross-sectional design of this study does not allow for strong directional causal inferences between COM factors and behavior to be determined [97]. Finally, while participants' demographic backgrounds appeared broadly consistent with NZ cat owners [1], the present study relied on an online panel sample which may not be generalizable to the wider population of NZ cat owners.

While we applied the COM-B model of behavior in our study [30], it is important to acknowledge that many behavior change theories exist [35]. However, from the perspective of behavior change practitioners and consistent with CBSM [29], we believe that the most effective way to initiate positive change involves reducing capability and opportunity barriers and increasing motivation. Furthermore, many behavior change models can be easily decomposed into COM-B [36] and given that this model is embedded within the BCW, it a practical framework to guide behavior change in real world settings.

Future research using a longitudinal research design with intensive measurement of behavior could be conducted to better understand the direction of the relationship between the COM factors and behavior, and to prevent recall bias. Additionally, research could explore whether cat owner segments differ in the characteristics of the owned cat (e.g., breed, sex, health status, temperament, previous access to the outdoors, behavioral issues). Finally, research should design interventions using a tool such as the BCW [30] to directly address the set of COM factors found to predict the cat containment behavior of the different cat owner audiences, in congruence with CBSM [29] and the behavior change process for animal management previously outlined [29, 31, 32]. Experimental designs with long-term follow ups can be used to assess the efficacy and effectiveness of interventions in promoting enduring behavior change in cat owners.

In conclusion, this study demonstrated that a sample of NZ cat owners are largely non-participants in cat containment, with various capability, opportunity, and motivational factors predicting their cat containment intentions and behavior. This study extends our understanding of cat containment behavior by supporting the influence situational and contextual factors have on whether NZ cat owners contain their cats. Audience segmentation analysis revealed that cat-owners surveyed were not a homogenous group, with four segments identified that have unique sets of COM factors predicting cat containment. More than three quarters were either *ambivalent* or *opposed* to containment, with only a minority being *engaged* or *receptive* towards it. Interventions seeking to promote cat containment among NZ cat owners should target those *receptive* and *ambivalent* towards cat containment to have the greatest overall impact on the issue of free-roaming companion cats.

## Supporting information

**S1 Appendix. Cat containment questionnaire.**
(PDF)

**S1 Table. Capability, opportunity and motivation variables and survey items.**
(DOCX)

**S2 Table. Summary of descriptive statistics for the four cat owner profiles for the capability, opportunity and motivations (COM) factors and demographics.**
(DOCX)

## Acknowledgments

We thank all the participants in this study for the time and input they provided. A special thanks as well to Andreas K. Jaeger for his support and proof-reading.

## Author Contributions

**Conceptualization:** Sarah A. E. Chamberlain, Donald W. Hine.

**Data curation:** Sarah A. E. Chamberlain.

**Formal analysis:** Sarah A. E. Chamberlain, Donald W. Hine.

**Investigation:** Sarah A. E. Chamberlain.

**Methodology:** Sarah A. E. Chamberlain, Lynette J. McLeod, Donald W. Hine.

**Supervision:** Donald W. Hine.

**Visualization:** Sarah A. E. Chamberlain.

**Writing – original draft:** Sarah A. E. Chamberlain.

**Writing – review & editing:** Sarah A. E. Chamberlain, Lynette J. McLeod, Donald W. Hine.

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
