## [Decision Letter · Decision Letter 0]

18 Sep 2023

PONE-D-23-20541Audience segmentation of New Zealand cat owners: Understanding the barriers and drivers of cat containment behaviorPLOS ONE

Dear Dr. Chamberlain,

Thank you for submitting your manuscript to PLOS ONE. After careful consideration, we feel that it has merit but does not fully meet PLOS ONE’s publication criteria as it currently stands. Therefore, we invite you to submit a revised version of the manuscript that addresses the points raised during the review process as noted below.

We look forward to receiving your revised manuscript.

Kind regards,

Christopher A. Lepczyk

Academic Editor

PLOS ONE

Additional Editor Comments:

Associate Editor:

Overall, this is an interesting and relevant manuscript on understanding cat containment. Both reviewers and I concur that the research is well done and provides a useful addition to the literature. However, there are a number of items that the reviewers and I have noted that need to be addressed in a revision.

Specific Comments

Throughout the ms, please add zeros before decimal points.

L30. This is a method, please move it to after you present the main goal/question of the ms.

L52. Please provide Latin name (species name) after cat.

L56. Please make sure to again state this is NZ as this is not the case in other nations.

L59. Please remove ‘less understood’ as while it may be relative to predation, we have very good knowledge about a number of diseases that they interact with and their ecological effects in terms of landscapes of fear, competition, etc.

L62. I would suggest noting the terrestrial species here even more than the marine as well as the fact that there are no animals in which we have looked that T. gondii has not been found. See Aguirre et al. Ecohealth 16:378–390. https://doi.org/10.1007/s10393-019-01405-7.

L66. I would suggest removing subheaders in the Intro and Discussion as they generally shouldn’t be needed.

L69, L88. What does ‘it’ mean? Replace as there is no object after ‘it’ to logically connect back to the previous sentence. Revise throughout the ms.

L182. Avoid beginning sentences with ‘this’ unless there is an object following that directly connects back to previous sentence.

L209. The main goal and the hypotheses should be included in the Abstract.

L222. Given that there are biases with such online marketing recruitment of surveys, please discuss why this method was used, known biases of the method, etc. I have some concerns of the type of individuals that may recruit into this survey vs. a random draw of cat owners.

L259. Avoid single sentence paragraph. You can either integrate this into the next paragraph or cut.

L333. Change ‘&’ to ‘and.’

L351. Rather than telling reader to see Table 1, describe the information in Table 1 and then put the table reference in parentheses.

L363. Same issue as previous point. Tell the reader about the information in a table or figure, then use parentheses to denote where it is found.

L367. Please indicate in Methods what you are using as the cutoff value for significance.

L369. Change ‘;’ to ‘:’

Table 1. Please make all figure and table legends as stand alone text. I wouldn’t be able to understand what this table is telling the reader without reading the text. Also, what does M mean? Also, why indicate a p-value is below 0.001? I would suggest you indicate if a value is below your cutoff value or give the exact value in the text. Finally, if values are correlated above 0.5, did you either remove one of them from modeling efforts or not use both in a model? I may have missed this point, but if you didn’t account for collinearity in models, then it is something that needs to be addressed.

Tables 2 and 3. No need for different p levels with asterisks. You either have a significant p-value or not. Please remove and simply indicate if a value met significance.

P19. Your line numbering stopped a few pages before this point. But again, here you note Table 4 just as a sentence. Please describe what the results in that table mean as with the other table points above.

P20. Same issue here with Fig. 1.

Table 4. Please show all models that were run, the delta AIC in scores and BIC, model weights, etc. Also, need to indicate in the Methods how top or best models were selected (was it delta AIC <2?), why or why not you did model averaging, etc. Also, I rarely see AIC and BIC used together. I did not see a good justification in the Methods why you mainly use frequentist statistics throughout and then use Bayesian for one set of analyses. Why do the Bayesian analyses on top of AIC?

P21. Change the following sentence as with other table/figure items above “Unstandardized segment descriptive statistics for all study variables are shown in S3 Table.”

P21-22. What type of post hoc tests were used (state in Methods) and what was the corrected cutoff values considered significant?

Discussion. Please cut first two paragraphs and start with the paragraph on page 24. Also, please remove subheaders from Discussion.

P23. Please delete the sentence “These findings are explored below and practical implications, recommendations for future research, and acknowledgements of the limitations of this study are provided.” as you don’t need to tell readers what you are going to say a few lines below.

P24. “Although Physical and Social Opportunity to Contain and Pro-Containment Beliefs were not significant predictors over-and-above the other factors, they were each independently associated with intentions and behavior.” Ok, but this gets at the issue of collinearity noted earlier. If these aren’t significant or other factors explain the data, then why discuss these? I would suggest cutting this information.

P25. I would also argue that the Crowley et al. ms used a different method, had low sample size, and in many ways was not a strong study. While you can cite it, I think the way they used Q method had a lot of limitations that your study does not.

P28. I would cut ‘Conclusion’ subheader and just have this be your final paragraph.

Figure 1. Please add in black x and y axes lines. I would also suggest changing fill patterns or color as the legend on the right is unclear as to what each bar represents.

Figure 2. Please add black y and x axes lines. Change y axes label to Mean. Also, can just state color of bars and what they mean in the legend.

Reviewers' comments:

Reviewer's Responses to Questions

**Comments to the Author**

1. Is the manuscript technically sound, and do the data support the conclusions?

Reviewer #1: Yes

Reviewer #2: Yes

2. Has the statistical analysis been performed appropriately and rigorously? 

Reviewer #1: Yes

Reviewer #2: Yes

3. Have the authors made all data underlying the findings in their manuscript fully available?

Reviewer #1: Yes

Reviewer #2: Yes

4. Is the manuscript presented in an intelligible fashion and written in standard English?

Reviewer #1: Yes

Reviewer #2: Yes

5. Review Comments to the Author

Reviewer #1: PONE-D-23-20541

Managing pet cats involves a challenging balance of animal welfare, community expectations and wildlife conservation, with solutions likely varying depending on cultural and ethnic factors. While focusing on the situation in NZ, the approaches described in this paper are broadly applicable in other jurisdictions with differing cultural norms regarding cat husbandry and interactions between cats and wildlife. The work is described clearly, with data collection, analysis and interpretation thorough and rigorous. Minor suggestions for improvement follow.

Introduction and discussion ¬– I share the authors’ view that responsible cat ownership, which I take to mean husbanding a cat in the interests of its welfare as well as that of the environment and human community, involves containing the cat on the owner’s property. That said, there are other authors arguing in the peer-reviewed literature that owners should allow their cats outdoor access, or that effects on wildlife are inconsequential (including papers with a NZ context). Ignoring this literature exposes one to charges of selectivity or bias. The solution is, I think, is to reference succinctly the concept of responsible ownership, acknowledge the critics, and then wrap with clear statements from animal welfare groups endorsing containment of pet cats. Relevant references for the different parts of this approach are:

Responsible ownership

Dalais RJ, Calver MC, Farnworth MJ (2023) Piloting an international comparison of readily accessible online English language advice surrounding responsible cat ownership. Animals 13, 2434. https://doi.org/10.3390/ani13152434.

Gunaseelan S, Coleman GJ, Toukhsati SR (2013) Attitudes toward responsible pet ownership behaviors in Singaporean cat owners. Anthrozoos 26, 198-211.

Elliott A, Howell TJ, McLeod EM, Bennett PC (2019) Perceptions of responsible cat ownership behaviors among a convenience sample of Australians. Animals 9, 703.

10.3390/ani9090703

Critics of containment or effects on wildlife

Abbate C (2021) Re-defending Feline Liberty: a Response to Fischer. Acta Analytica 36, 451-463.

Abbate CE (2020) A Defense of Free-Roaming Cats from a Hedonist Account of Feline Well-being. Acta Analytica 35, 439-461.

Flux JEC (2007) Seventeen years of predation by one suburban cat in New Zealand. New Zealand Journal of Zoology 34, 289-296.

Flux JEC (2017) Comparison of predation by two suburban cats in New Zealand. European Journal of Ecology 3, 85-90.

Welfare groups endorsing containment

RSPCA Australia Policy A09 Cat management, section 9.4 – https://kb.rspca.org.au/knowledge-base/rspca-policy-a09-cat-management/

SPCA Policy Brief on National Cat Legislation for New Zealand: Background document, page 5 – https://www.spca.nz/images/assets/772883/1/national%20cat%20legislation_bd_v2_final.pdf

Line 55 – isn’t it the perceived friendship, love and affection they offer? This might not be the way the cat understands the situation.

Line 61 – Toxoplasma gondii, not toxoplasmosis gondii.

Lines 62-63 – this may be the place to acknowledge critics of cat containment.

Line 74 – ‘performance of cat containment.’ Would ‘incidence’ or ‘prevalence’ be better?

Lines 111-112 – ‘Opportunity is comprised of physical and social opportunity and are external factors ...’ Would ‘that are external factors’ be better?

Line 154 and elsewhere – The abbreviation AU is used for Australia. I don’t think this abbreviation is used widely elsewhere, so Australia in full may be better.

Line 347 – what version of SPSS was used?

Reviewer #2: This manuscript describes an audience research study to assess potential barriers and drivers of participation in cat containment for New Zealand cat owners. They found four distinct segments of cat owners with markedly different profiles related to capability, opportunity, and motivation to perform cat containment. In addition, the four groups differed significantly in their cat containment intentions and behaviors. I appreciated their inclusiveness for what was considered cat containment, moving away from the over-simplified “cats indoors” focus. It also provided important insights into contextual differences between countries, which has important implications for intervention design and expectations for uptake. The research was well-grounded in previous cat owner behavior studies and the survey was designed well. The paper was also well written and easy to follow. I only had minor comments, which mostly related to opportunities to engage with the broader behavior change literature.

In the introduction, there is brief mention of a couple of well-known approaches to behavior change, but the growth of the field and theoretical background is not really described. Much of the focus is on behavior change studies conducted on cat owners, which is helpful, but does not give an accurate image of how large and established behavior change is as a discipline (albeit not usually applied to natural resources conservation – although that is growing). For example, the Society for Conservation Biology now has a conservation marketing working group, there are targeted trainings for conservation biologists that are regularly offered by a number of groups (beyond Doug McKenzie-Mohr), the Social Marketing Association of North America has started engaging with conservation groups (largely through the SCB working group’s Conservation Marketing conferences), etc. And this is not including other approaches to behavior change that can be applied, beyond conservation marketing. It would also help you situate your selection of the COM-B model from the other behavior change approaches you could have selected. What other ones did you consider and why was COM-B considered the best one to use? I also noted a technical term that could have used some explanation (injunctive social norm). Again, with a more robust introduction section, this could easily be covered.

Similarly, in the discussion, it would be helpful to link your findings back to behavior change theory. The most obvious one to me is the very different COM factors for the audience segments, compared to what looks like a simple gradation of behaviors and intentions – if you want more behavior, why not just use more of the same interventions. Discussing why theory tells you this would be a flawed approach would be helpful. What do principles of behavior change design interventions (and even strategic communication in general) say about how to think about different audiences? And importantly, how do you design your intervention to be sure it reaches the target audience segment, vs. all cat owners generally? You do have some citations in the discussion, but again they rely heavily on the COM-B framework and previous cat studies. There is a much broader range of literature that would support many of your findings and strengthen your arguments.

It would also be good to consider the ethics of when you are promoting a change in behavior. Who decides what behavior is desired and what are the implications of that? Especially if you're using social psychology and other frameworks that some might consider "social engineering". This should at least be mentioned somewhere, either in the intro or discussion or both.

Overall, this paper was a pleasure to read and provides insights that will help advance global considerations of how to address one source of outdoor cats. In addition to the comments above, I have attached a pdf with more specific comments using the pdf highlight and comment options. With attention to these minor edits, the paper will be a nice contribution to the growing body of behavior change and cat owner literature.

6. PLOS authors have the option to publish the peer review history of their article (what does this mean?). If published, this will include your full peer review and any attached files.

Reviewer #1: No

Reviewer #2: **Yes: **Kirsten Leong

---

## [Author Response · Author response to Decision Letter 0]

30 Nov 2023

Please see the attached response to reviewers document for a response to all comments provided.

---

## [Editor Report · Decision Letter 1]

19 Dec 2023

Audience segmentation of New Zealand cat owners: Understanding the barriers and drivers of cat containment behavior

PONE-D-23-20541R1

Dear Dr. Chamberlain,

We’re pleased to inform you that your manuscript has been judged scientifically suitable for publication and will be formally accepted for publication once it meets all outstanding technical requirements.

Kind regards,

Christopher A. Lepczyk

Academic Editor

PLOS ONE
---

## [Editor Report · Acceptance letter]

2 Jan 2024

PONE-D-23-20541R1 

PLOS ONE

Dear Dr. Chamberlain, 

I'm pleased to inform you that your manuscript has been deemed suitable for publication in PLOS ONE. Congratulations! Your manuscript is now being handed over to our production team.

Kind regards, 

on behalf of

Dr. Christopher A. Lepczyk 

Academic Editor

PLOS ONE